# Bioactive Metabolites from Marine Algae as Potent Pharmacophores against Oxidative Stress-Associated Human Diseases: A Comprehensive Review

**DOI:** 10.3390/molecules26010037

**Published:** 2020-12-23

**Authors:** Biswajita Pradhan, Rabindra Nayak, Srimanta Patra, Bimal Prasad Jit, Andrea Ragusa, Mrutyunjay Jena

**Affiliations:** 1Algal Biotechnology and Molecular Systematic Laboratory, Post Graduate Department of Botany, Berhampur University, Brahmapur 760007, India; pradhan.biswajita2014@gmail.com (B.P.); rabindran335@gmal.com (R.N.); 2Cancer and Cell Death Laboratory, Department of Life Science, National Institute of Technology Rourkela, Rourkela 769001, India; 518LS2007@nitrkl.ac.in; 3Department of Biochemistry, All India Institute of Medical Science, Ansari Nagar, New Delhi 110023, India; bimaljit2019@gmail.com; 4Department of Biological and Environmental Sciences and Technologies, Campus Ecotekne, University of Salento, via Monteroni, 73100 Lecce, Italy; 5CNR-Nanotec, Institute of Nanotechnology, via Monteroni, 73100 Lecce, Italy

**Keywords:** marine bioactive compounds, secondary metabolites, algae, oxidative stress, ROS, cancer, diabetes, inflammation, apoptosis

## Abstract

In addition to cancer and diabetes, inflammatory and ROS-related diseases represent one of the major health problems worldwide. Currently, several synthetic drugs are used to reduce oxidative stress; nevertheless, these approaches often have side effects. Therefore, to overcome these issues, the search for alternative therapies has gained importance in recent times. Natural bioactive compounds have represented, and they still do, an important source of drugs with high therapeutic efficacy. In the “synthetic” era, terrestrial and aquatic photosynthetic organisms have been shown to be an essential source of natural compounds, some of which might play a leading role in pharmaceutical drug development. Marine organisms constitute nearly half of the worldwide biodiversity. In the marine environment, algae, seaweeds, and seagrasses are the first reported sources of marine natural products for discovering novel pharmacophores. The algal bioactive compounds are a potential source of novel antioxidant and anticancer (through modulation of the cell cycle, metastasis, and apoptosis) compounds. Secondary metabolites in marine Algae, such as phenolic acids, flavonoids, and tannins, could have great therapeutic implications against several diseases. In this context, this review focuses on the diversity of functional compounds extracted from algae and their potential beneficial effects in fighting cancer, diabetes, and inflammatory diseases.

## 1. Introduction

Epidemiological studies have evidenced the dangerous effects on human health of the ever-increasing intake of junk food, alcohol, and antibiotics. This bad behavior can increase the risk of oxidative stress which, in turn, can lead to accelerated aging and inflammatory diseases, such as cardiovascular and neurodegenerative disease and many types of cancer [1]. According to reports by the WHO, more than 200 types of lethal cancer accounted approximately for 9.6 million deaths per year in 2019 globally [2]. Similarly, diabetes mellitus, a metabolic disorder, has emerged as the third foremost cause of death worldwide (1.6 million deaths per year) with several associated ill-fated diseases, such as heart attack, stroke, kidney failure, high blood pressure, blindness, and lower limb amputation [3,4,5]. According to WHO, the world’s diabetic population will hike up to 592 million by 2035. In addition to cancer and diabetes, inflammatory diseases have tremendously increased in the recent past causing millions of deaths [6]. Unfortunately, current chemotherapeutic, anti-diabetic, and anti-inflammatory drugs often present several adverse effects, such as toxicity, drug tolerance, and metabolic impairments [1]. In this regard, natural products might provide alternative drugs with better characteristics [7]. Similarly, their regular uptake through diet or novel pharmacological formulations might help prevent oxidative stress-related diseases [8,9,10,11].

Approximately 70% of the Earth’s surface is covered by oceans and it hosts an immense variety of marine organisms which represent a rich source of natural products [1,12,13]. Marine algae are among the most promising sources of novel bioactive compounds with interesting biological effects, such as antioxidant, anticancer, antibacterial, antifungal, antidiabetic, and anti-inflammatory activities [1]. Marine algae are extensively used in diet and traditional medicine in Asian countries because of the presence of minerals, dietary fiber, lipids, omega-3 fatty acids, proteins, polysaccharides, and essential amino acids [14,15]. They also contain many vitamins, such as vitamins A, B, C, and E [16]. Few marine algae-derived bioactive compounds, such as phlorotannins, polysaccharides, fucoidans, alginic acid, tripeptides, pyropheophytin, and oxylipin, have been shown to reduce the risk of cancer, diabetes, and inflammatory diseases [15]. Hence, in this review we focused our attention on the diversity of marine algal bioactive compounds and the recent findings about their molecular mode of action in potentially fighting cancer, diabetes, and inflammation (Figure 1). Furthermore, algal extracts showed potential antimicrobial activity against aerobes, psychotropic, proteolytic, and lipolytic bacteria and act as natural preservatives. Additionally, they can prevent lipid oxidation [17,18].

## 2. Biological Activities of Marine Algae and Potential Health Benefits via Dietary Supplements

Diet plays an important role in disease prevention as more than 33% of diseases, such as cancer, diabetes, and inflammation-associated chronic diseases, could be avoided by changing lifestyle and food habits [19,20]. Nutritional supplements from natural sources could also play an important role in preventing diseases. Phytochemicals from marine algae, such as peptides, amino acids, lipids, fatty acids, sterols, polysaccharides, carbohydrates, polyphenols, photosynthetic pigments, vitamins, and minerals, some of which are represented in Figure 2, can act as potent antioxidants and have beneficial effects as anti-diabetic and chemotherapeutic drugs, as detailed below.

### 2.1. Peptides and Amino Acids

Hydrolysis of proteins can lead to bioactive peptides that can present beneficial health aspects and modulate the outcome of the disease. Bioactive phyto-peptides have 3 to 20 amino acid residues and display biological properties such as antioxidant, anticancer, anti-inflammation, and immunomodulation. For example, purified peptides from *Chlorella vulgaris* can prevent cellular damage and can act as potent anticancer agents [21,22].

The protein content in macro and micro algae comprises all essential amino acids which prevent cellular damage. The red alga *Palmaria palmata* is rich in Leu, Val, and Met, and their mean levels are similar to ovalbumin. Similarly, Ile and Thr concentrations are comparable to those in legume proteins. The green alga *Ulva rigida* contains Leu, Phe, and Val as major essential amino acids [23,24].

### 2.2. Lipids and Fatty Acids

The structural complexity of lipids and fatty acids are highly diverse and contribute to their therapeutic efficacy. It has been reported that small amounts of saturated fatty acids can help prevent cardiovascular diseases. Marine algae contain polyunsaturated fatty acids (PUFAs) and significant amounts of monounsaturated fatty acids which are beneficial to human health and could help reduce cardiovascular diseases [25].

#### 2.2.1. Polyunsaturated Fatty Acids (PUFAs)

Humans are incapable of synthesizing PUFAs that, on the other hand, are abundant in both macro and microalgae. PUFAs in microalgae are mainly composed of omega-3 and omega-6 fatty acids (e.g., EPA and AA) [26]. PUFAs regulate blood clotting and blood pressure and modulate the function of the brain and nervous systems [27]. Moreover, they decrease the risk of several chronic diseases, such as diabetes and cancer. Additionally, they regulate inflammatory responses by producing eicosanoids, well-known inflammation mediators [27]. Omega-3 and omega-6 PUFA from macroalgae are already used as dietary supplements [28]. Red and brown algae have a high level of omega-3 fatty acids (e.g., EPA and GLA) and omega-6 fatty acids (e.g., AA and linoleic acid). Brown algae *Laminaria ochroleuca* and *Undaria pinnatifida* are a rich source of octadecatetraenoic acid, an omega-3 PUFA [29]. Green seaweed *Ulva pertusa* was found to be rich in hexadecatetraenoic (omega-3), oleic (omega-9), and palmitic acids (SFA). The lipid fraction of microalgae *C. vulgaris* contains oleic, palmitic, and linolenic acids. The green microalga *Haematococcus* sp. also contains short-chain fatty acids. Long-chain PUFAs are also used as nutritional supplements and food additives [30]. *Spirulina* sp. is a promising source of GLAs, a precursor of leukotrienes, prostaglandins, and thromboxans that regulate inflammatory, immunological, and cardiovascular disorders. Cyanobacteria and some green algae also contain bioactive fatty acids such as palmitic, oleic, and lauric acids along with DHA [22].

#### 2.2.2. Sterols

Sterols are a class of lipids extensively found in both macro and microalgae. Sterols and some of their derivatives have potential biological, e.g., anti-inflammatory, activity. Sterols from *Spirulina* triggers the formation of the plasminogen-activating factor in vascular endothelial cells. Fucosterol, ergosterol, and chondrillasterol are found in brown algae and cholesterol has been found in red algae [22,31].

### 2.3. Polysaccharides and Carbohydrates

Polysaccharides are abundant in seaweeds and also found in microalgae. They generally comprise about 4% to 76% of the total dry weight of the alga. Polysaccharides are classified according to their chemical structure, such as sulfuric acid polysaccharides, sulfated xylans, and galactans (generally found in green algae). Moreover, alginic acid, fucoidan, laminarin, and sargassan are found in brown algae [32]. Agar, carrageenans, xylans, and floridean are generally found in red algae. Many algal polysaccharides present bioactivity and could become drug candidates with potential use in several human health disparities [33]. Carrageenans are sulfated galactans and they are extensively used in pharmaceutical and food industries. Soluble fibers such as fucans, alginates, and laminarans are found in brown seaweeds, whereas soluble fibers such as sulfated galactans (agars and carrageenans), xylans, and floridean starch are abundantly found in red seaweeds [34]. Green algae contain xylans, mannans, starch, and ionic sulfate group-containing polysaccharides in combination with uronic acids, rhamnose, xylose, galactose, and arabinose. Many of the polysaccharides can be regarded as dietary fibers and are classified into two groups, i.e., soluble and insoluble fibers [35,36]. Seaweeds contain about 25% to 75% dietary fibers in comparison to their dry weight, a higher percentage compared to that found in fruit and vegetables [37]. The algal dietary fiber consumption has several health benefits as they can be used as antitumor, anticancer, anticoagulant, and antiviral agents. Fucoidans are extensively found in the cell walls of brown macroalgae [38]. Fucoidans have several biological activities and act as antioxidant, antitumor, anti-inflammatory, antidiabetes, antiviral, anticoagulant, and antithrombotic agents. Additionally, they also modulate the human immune system [1]. Furthermore, laminarin is the second main source of glucan, abundantly found in brown algae, and it acts as a facilitator of intestinal metabolism [36].

Carbohydrates, such as glucose and starch, are abundantly found in microalgae [39]. Many biological functions of microalgal species are due to the presence of carbohydrates. *Chlorella pyrenoidosa* and *Chlorella ellipsoidea* contain glucose and a wide variety of combinations of galactose, mannose, rhamnose, *N*-acetylglucosamine, *N*-acetylgalactosamine, and arabinose that exert immune-modulatory and antiproliferative activity [40]. β-1,3-Glucans extracted from *Chlorella* have been shown to act as immunomodulators that can reduce blood lipids [41].

### 2.4. Polyphenolic Compounds

Marine algal bioactive compounds are potent antioxidant agents that protect from oxidative damage [42]. The antioxidant activity of bioactive compounds from marine algae is associated with protection against cancer, inflammatory, diabetes, and several ROS-related diseases [43]. Polyphenolic compounds are mainly found in both micro and macroalgae [42]. The phenolic components include hydroxycinnamic acids, phenolic acids, simple phenols, xanthones, coumarins, naphthoquinones, stilbenes, flavonoids, anthraquinones, and lignins [21]. The phlorotannins with potential antioxidant activity belong to polyphenolic compounds that have been screened from several brown algae. Phlorotannins are known for their chemopreventive, antibacterial, antiproliferative, and UV-protective properties [22].

### 2.5. Photosynthetic Pigments

Macroalgae contain chlorophylls and carotenoids as major photosynthetic pigments. Carotenoids are well known for their antioxidant properties and dietary carotenoids have high nutritional and therapeutic value [44]. Carotenoids are well known for their chemopreventive effect against several cancer subtypes. Microalgae are also the main source of antioxidants such as β-carotene and astaxanthin. β-Carotene is a natural colorant that has been conventionally used as food and drinks colorants and can act as dietary food supplements or additives with a high antioxidant capacity [45].

### 2.6. Vitamins and Minerals

Vitamins are micronutrients essential for human body growth and development. Seaweeds and microalgae are known to be a good source of vitamin B1, B2, and B12. Vitamin B12 (cobalamin) is extensively found in higher concentrations in green and red algae compared to brown algae [46]. Vitamin B12 is generally found in red macroalgae such as *Palmaria longat* and *Porphyra tenera*. The highest vitamin B12 content was found in red seaweed *Porphyra* sp. and green algae, such as *Enteromorpha* sp. and *Spirulina*. Cobalamin deficiency can cause health diseases, such as neuropsychiatric disorders and megaloblastic anaemia. Vitamin C (ascorbic acid) is present in all red, brown, and green seaweeds. Vitamin C has several health benefits, such as radical scavenging activity, antiaging, and immune stimulant activity. Vitamin E is a mixture of tocopherols. α-Tocopherol occurs in green, red, and brown seaweeds. Phaeophyceae also contain β- and γ-tocopherols and displayed outstanding *antioxidant* activity. Vitamins C and E were also found in *Laminaria digitata* and *U. pinnatifida* [22].

Seaweeds and macroalgae are rich in minerals, trace elements, and maintain inorganic atoms in seawater. Minerals and trace elements are required for the human diet [47]. Phaeophyceae, such as *U. pinnatifida* and *Sargassum*, and rhodophyta, such as *Chondrus crispus* and *Gracilariopsis*, are considered dietary supplement that meet the recommended daily intake of some of the major minerals, such as Na, K, Ca, and Mg, as well as trace minerals, such as Fe, Zn, Mn, and Cu. In addition, seaweeds are also important sources of Ca as they reduce Ca deficiency risk in pregnant women and adolescents and they inhibit preadolescent aging [48].

## 3. Marine Bioactive Metabolites and Their Therapeutic Efficacy

The marine ecosystem is a source of novel natural secondary metabolites with promising biomedical applications [49]. The impact of marine algae in the area of traditional medicine is huge and they have been used as *Yunani hakim* in many countries, such as China and Egypt. Marine algae produce diverse secondary metabolites and might be the most promising sources of proteins, vitamins, omega-3, carotenoids, phenolic acids, and flavonoids, as well as other natural antioxidants [50]. These marine bioactive compounds act as free radical scavengers and prevent oxyradical formation thus reducing oxidative stress and, as such, they have great importance in the prevention of cancer, diabetes, early aging, and several other inflammatory diseases (Figure 3).

Marine bioactive compounds, such as algal photosynthetic pigments, phycobiliproteins, carotenoids, polyphenols, terpenes, phlorotannins, and polysaccharides, have shown promising therapeutic activity in both in vitro and in vivo models [25,29,38,51].

### 3.1. Marine Bioactive Metabolites and Modulation of In Vitro Antioxidant Activity

Reactive oxygen species (ROS) comprise a group of oxygenic ions that are highly reactive and pose a serious threat to biological components, ultimately leading to serious disorders such as cancer, diabetes mellitus, neurodegenerative and inflammatory diseases [52]. The oxygen-containing radicals comprise of peroxyl (ROO^•^), hydroxyl (OH^•^), hydroperoxyl (HO_2_^•^), superoxide (O_2_^•^), alkoxyl (RO^•^), thiyl peroxyl (RSOO^•^), sulfonyl (ROS^•^), and nitric oxide (NO^•^) radical, as well as non-radical oxidizing agents such as singlet oxygen (^1^O_2_), hydrogen peroxide (H_2_O_2_), hypochlorous acid (HOCl), and organic hydroperoxides (ROOH) [52,53,54,55]. Cells can detoxify ROS as they are furnished with antioxidant defense mechanisms to maintain cellular equilibrium. Antioxidants fight against ROS and exert a positive effect on human health by protecting macromolecules such as proteins, DNA, and membrane lipids [56]. The use of synthetic antioxidants used as food additives, such as butylated hydroxyanisole, butylated hydroxytoluene, tertiary-butylhydroquinone, and propyl gallate, might represent a threat because of their side effects [57]. Hence, the development of novel antioxidants from natural sources like marine flora can represent a promising approach. Marine algae could neutralize ROS because of their antioxidant compounds, such as phycobilins, phycocyanin, carotenoids, astaxanthin, polyphenols, and vitamins, which can act against cancer, diabetes, inflammation, aging, and immune responses. The antioxidant capacity of various marine algae, such as green, red, and brown algae species, has been already extensively reported in the literature.

Antioxidant activity of marine algal compounds have been determined by several methods, such as 2,2-diphenyl-1-picrylhydrazil (DPPH) radical scavenging, ferric reducing antioxidant power (FRAP), lipid peroxide inhibition, ABTS radical scavenging, nitric oxide (NO) scavenging, hydrogen peroxide radical scavenging assays, superoxide radical and hydroxyl radical scavenging assays. The methanolic extract of blue-green algae has shown potent DPPH radical scavenging activity. In addition, phycocyanin from *Spirulina platensis* showed strong H_2_O_2_ scavenging activity [58]. Antioxidant properties in green algae *Ulva fasciata* and *Ulva reticulate* were characterized by free-radical-scavenging due to the presence of flavonoids [59,60,61]. In brown algae such as *Ecklonia cava*, *Eisenia bicyclis*, and *Ecklonia kurome* antioxidant activities were characterized by DPPH-radical scavenging [62]. Ethanolic extracts of *Gracilaria tenuistipitata* and *Callophyllis japonica* also have shown potential antioxidant activities [63,64].

### 3.2. Intricate Role of Algal Bioactive Metabolites as Anticancer Agents

Free radicals and ROS generally promote cancer initiation. Synthetic chemopreventive drugs often present several adverse side-effects to the tumor vicinity and bodily organs because of poor specificity and generalized biodistribution [65]. Several marine algal bioactive compounds have been designated as potent chemopreventives due to inhibition of cellular proliferation, modulation of the cell cycle, and induction of apoptosis [66,67].

#### 3.2.1. Inhibition of Cell Proliferation

Several studies have reported that marine algal bioactive compounds have antiproliferative and inhibitory activity against several cancer subtypes in in vitro as well as in vivo [68]. Sulfated polysaccharides purified from brown seaweeds exhibited an antiproliferative effect on human leukemia and lymphoma cell lines [38]. They have also been reported to inhibit proliferation of breast (MCF-7) and cervical (HeLa) cancer cells [38]. Sulfated polysaccharides extracted from the brown seaweed *Sargassum vulgare* displayed inhibition of cell proliferation in HeLa and B16 cells without cytotoxicity in normal rabbit aortic endothelial cells [69]. Furthermore, sulfated polysaccharides from red seaweed *Amansia multifidi* inhibited the cellular viability of HeLa cells. The polysaccharides isolated from *Gracilariopsis lemaneiformis*, consisting of 3,6-anhydro-L-galactose and D-galactose and a linear structure of repeated disaccharide agarobiose units, hindered the viability of B16, A549, and MKN-28 cell lines [70]. Similarly, low-molecular-weight sulfated polysaccharides from green seaweed *Gayralia oxysperma* inhibited the cell viability of U87MG glioblastoma cells even at microgram-level concentrations (10, 100, and 1000 µg/mL) without any evident cytotoxicity [71]. Fucoidans isolated from *Undaria pinnatifida* also showed inhibition of cellular viability in SK-MEL-28, T-47, RPMI-7951, T47D, and DLD-1 cancer cell lines at microgram concentrations. Moreover, fucoidans isolated from the sporophyll of *U. pinnatifida* demonstrated to be able to inhibit cell growth in HeLa, A549, PC-3, and HepG2 cell lines, although at higher concentrations (treatment with 0.8 mg/mL for 24 h) [72]. Furthermore, in prostate cell lines (DU-145), fucoidans treatment marked a 90% reduction in cell viability [73]. Similarly, *Fucus vesiculosus* derived fucoidans were reported to reduce cell viability of human colorectal carcinoma (HCT116) cell line by 60% [74]. Fucoidan from *Ecklonia cava* have been also reported to inhibit proliferation of MDA-MB-231 cells. Moreover, the administration of fucoidan (20 mg/kg for 28 days) in a DU-145 cell-induced xenograft rat model has reduced the tumor growth by 50% [73]. Carrageenans from *Kappaphycus alvarezii* reduced the growth of liver, colon, breast, and osteosarcoma cell lines [75]. Similarly, treatment of phlorotannins (a type of polyphenol) isolated from *E. cava* has marked a reduction in cell viability in MDA-MB-231 and MCF-7 cells by 55% and 64%, respectively [76]. Similarly, phlorotannin-rich extracts from *Ascophyllum nodosum* reduced the cell viability of HT-29 colon cancer cells [77].

Halogenated monoterpenes isolated from red seaweeds *Plocamium cornutum* and *Plocamium suhrii* displayed potent antiproliferative activity as compared to anticancer drug cisplatin [78]. *U. pinnatifida* isolated fucoxanthins has a cytotoxic effect against LNCaP, DU145, PC-3, Caco-2, HT-29, DLD-1, HeLa, and Jurkat cell lines [79,80]. Moreover, fucoxanthinol displayed an anti-proliferative effect against drug-resistant HT-29-derived cells, and inhibited xenograft tumor development in a dose-dependent manner [81]. The guaiane sesquiterpene derivative guai-2-*en*-10-ol isolated from the green seaweed *Ulva fasciata,* reduced viability of breast cancer MDA-MB-231 cell line [82]. *G. tenuistipitata* aqueous extract counteracted the cellular proliferation in H1299 cells. Heterofucans from *Sargassum filipendula* exhibited anti-proliferative effects on cervical, prostate, and liver cancer cells [83]. Aqueous extracts of *Sargassum oligocystum* and *Gracilaria corticata* inhibited proliferation of human leukemic cell lines [84,85]. Ethanolic and methanolic extracts of *Gracilaria tenuistipitata* exhibited anti-proliferative effects against Ca9-22 oral cancer cells [86,87,88]. Several studies have reported that algae consumption modulates cancer prevention. The diets containing seaweeds decreased the growth of DU-145 human prostatic tumor cells in nude mice. Moreover, the administration of red algae *Eucheuma cottonii* extracts as dietary supplement to rats displayed tumor repression [89].

#### 3.2.2. Cell Cycle Arrest and Inhibition of Angiogenesis

Inhibition of cell cycle hinders cancer cell proliferation for the subsequent exhibition of anticancer activity. Sulfated polysaccharides from *G. oxysperma* arrested the cell cycle [71]. Fucoidan from *Fucus vesiculosus* arrested the cell cycle at the G1 phase in HCT116 human colorectal carcinoma and HT-29 colon cancer cells [74]. Fucoxanthin arrested the cell cycle via downregulation of cyclin D1, D2, CDK4 and upregulation of p15INK4B and p27Kip1 expression [90]. Fucoxanthin from *Laminaria japonica* arrested the sub-G1 phase of the cell cycle in WiDr cancer cells [91]. Moreover, in LNCap prostate cancer cells, fucoxanthin arrested the G1 phase of the cell cycle via MAPK/ JNK and GDD45A pathways [92]. Pheophorbide a, from *G. elliptica* arrested the cell cycle in the G0/G1 phase in glioblastoma cells [93]. Aqueous extract of *G. tenuistipitata* induced G2/M arrest in the H1299 cell line [64].

Angiogenesis plays a key role in tumor growth and metastasis. Polysaccharides isolated from *S. vulgare* exhibited angiogenesis inhibitory activity. Fucoidans isolated from *U. pinnatifida* significantly reduced the expression of the angiogenesis factors VEGF-A and VEGF-162 [94]. Sulfated polysaccharides from brown seaweed *Sargassum vulgare* displayed antiangiogenic activity in HeLa and B16 cells without damage to the tumor vicinity [69]. Furthermore, dieckol decreased the expression of angiogenic markers such as PCNA, VEGF, COX-2, MMP-2, and MMP-9 to inhibit metastasis [95].

#### 3.2.3. Induction of Apoptosis

Apoptosis (or programmed cell death, PCD) is the main goal of anticancer drugs. Several reports have demonstrated the role of algal bioactive compounds and polysaccharides as potent anticancer agents by modulating apoptosis, as schematized in Figure 4. Sulfated polysaccharides from *Phaeophyceae* act as novel chemopreventive drugs owing to their free-radical scavenging activity [1]. Sulfated polysaccharides induced apoptosis in human leukemic monocyte lymphoma cell line (U-937) [1]. Polysaccharides from *Capsosiphon fulvescens* induced apoptosis in gastric cancer cells via modulation of PI3K/Akt pathway [96]. Polysaccharides from *U. lactuca* increased the activity of antioxidant enzymes in a DMBA-induced breast cancer model via diminished lipid peroxidation as well as GSH-Px activity to restrain apoptosis [97]. Polysaccharides from red seaweed *Champia feldmannii* demonstrated in vivo antitumor effects in mice transplanted with sarcoma 180 tumors via modulation of apoptosis [98]. The polysaccharides isolated from sea lettuce *U. lactuca* displayed in vitro and in vivo anticancer activity in breast cancer via modulation of apoptosis. It also displayed a chemopreventive effect in DMBA-induced breast cancer in rat post-administration for 10 weeks and prevented breast-histological alterations and carcinogenic wounds. Additionally, it also amplified the p53 expression and inhibited the Bcl-2 expression to induce apoptosis in breast cancer cells [97].

Fucoidans isolated from the sporophyll of *U. pinnatifida* displayed apoptosis in DU-145 cell-induced xenograft rat model via inhibition of the JAK3/STAT pathway. In addition, fucoidan (IC_50_ 530 ± 3.32 mg/mL) also reduced the viability of B16 melanoma cells via activating apoptosis [99]. Fucoidan from *Fucus vesiculosus* induced p53-independent apoptosis in HCT116 human colorectal carcinoma cell line [74]. Fucoidan from *L. japonica* induced apoptosis via activation of caspase-3, poly(ADP-ribose) polymerase (PARP), and DNA degradation in HT-29 cell line [100]. Moreover, fucoidan from *E. cava* induced apoptosis in MDA-MB-231 and MCF-7 cells via induction of p53 and activation of Bax, caspases 3 and 9, and PARP with inhibition of Bcl-2 [76]. Furthermore, fucoidan from *Cladosiphon okamuranus* displayed induction of apoptosis in MCF-7 cells via activation of caspase-3 and DNA fragmentation [101]. *F. vesiculosus* extracts enhanced mitochondria membrane permeability thus inducing apoptosis via cytoplasmic release of cytochrome C and the Smac/DIABLO pathway in human colon cancer cells [102]. Similarly, fucoidan from *F. vesiculosus* treatment induced apoptosis in HT-29 colon cancer cells via decreasing the expression of Bcl-xL, Bcl-2 and upregulation of Bax, pro-caspases 3, 7, and 9. An upregulation of Rb and E2 factor proteins and Fas-regulated extrinsic apoptosis was also evident post fucoidan treatment in HT-29 colon cancer cells [103].

Fucoxanthin from *Laminaria japonica* induced apoptosis via DNA fragmentation in human colon adenocarcinoma WiDr cells [91]. Fucoxanthin from *Ishige okamurae* exhibited anticancer activity in melanoma B16F10 cells both in vitro (B16F10 cell line) and in vivo (Balb/c mice implanted with B16F10 cells) via induction of apoptosis. Moreover, fucoxanthin induced apoptosis via caspases activation and reduction of BclxL and IAP expression [90]. Laminarin from *Laminaria digitata* induced apoptosis in human colon cancer (HT-29) cells and activated ErbB2 phosphorylation. Moreover, it also inhibited cell proliferation and induced apoptosis in prostate cancer (PC-3) cells and increased the expression of P27kip1 and PTEN [104]. Dieckol from *E. cava* daily administration (40 mg/kg for 15 weeks) decreased cancer cell proliferation in albino rats via induction of apoptosis [95]. The guaiane sesquiterpene derivative guai-2-*en*-10-ol from green seaweed *Ulva fasciata* induced apoptosis in MDA-MB-231 breast cancer cell line via direct interaction with the kinase site of EGFR [82]. The halogenated monoterpene, mertensene from red alga *Pterocladiella capillacea* induced apoptosis in HT-29 and LS174 via modulation of ERK1/2, Akt, and NF B pathways [78]. Ethanolic and methanolic solvent extracts of *Gracilaria tenuistipitata* displayed apoptosis in Ca9-22 oral cancer cells via DNA damage. Furthermore, methanol extract of *Plocamium telfairiae* induced caspase-dependent apoptosis in HT-29 colon cancer cells [88]. Iodine and polyphenols from *L. japonica* induced apoptosis via inhibiting SOD activity [105]. The red alga *Porphyra yezoensis* can induce cancer cell death via apoptosis in a dose-dependent manner in in vitro cancer cell lines without exhibiting cytotoxicity towards the normal cells. Moreover, Carrageenans, heterofucans, dieckol, and iodine can induce cancer cell death via apoptosis in a dose-dependent manner in in vitro cancer cell lines without exhibiting cytotoxicity towards the normal cells. Furthermore, *L. japonica* water extracts induced apoptosis in several human breast cancer cell lines. Moreover, *Eucheuma cottonii* extract displayed the upregulation of antioxidant enzymes such as catalase (CAT), superoxide dismutase (SOD), glutathione peroxidase (GPx) in cancer-induced rats [89]. Methanolic extracts of *Fucus serratus* and *F. vesiculosus* exhibited protection of DNA damage induced by H_2_O_2_ in Caco-2 cells [106]. Furthermore, *Pelvetia canaliculata* inhibited H_2_O_2_-induced superoxide dismutase depletion in Caco-2 cells [106]. *C. japonica* ethanol extracts inhibited H_2_O_2_-induced apoptosis via activating cellular antioxidant enzymes [63]. *G. tenuistipitata* aqueous extract enhanced the recovery of these cells from H_2_O_2_-induced DNA damage in the H1299 cell line [59]. Apoptosis modulation by algal metabolites in different cancerous cell lines with molecular pathways are summarized in Table 1.

### 3.3. Anti-Inflammatory Activity of Marine Algal Bioactive Metabolites

Inflammation is a molecular marker of carcinogenesis. Marine natural products are well-known anti-inflammatory agents due to their potent antioxidant activity. Several anti-inflammatory compounds with potential pharmacological applications have been isolated from marine algal sources. Macroalgae contain several polysaccharides, such as fucoidan, fucans, alginates, laminarin, agar, and carrageenans, which are used as prebiotic compounds and that can have potential application as anti-inflammatory agents. Marine-derived carotenoids and astaxanthin exhibit potent anti-inflammatory activity [108,109].

The anti-inflammatory activity of marine algae is due to the presence of PUFAs (e.g., omega-3) that potentiate inhibition of inflammation [27]. Several studies have demonstrated that omega-3 to 6 fatty acids reduce inflammation when taken as dietary supplements [27]. The polysaccharide extracted from *Turbinaria ornate, Delesseria sanguinea* exhibited anti-inflammatory potential in several in vitro systems. Sulfated polysaccharide fraction from *Gracilaria caudate*, a galactan from *Gelidium crinale*, a mucin-binding agglutinin from *Hypnea cervicornis,* lectin from *Pterocladiella capillacea*, and sulfated galactofucan from *Lobophora variegata* also exhibited anti-inflammatory potency [110]. Oral administration of marine polysaccharide in an in vivo mouse model reduced the initiation of inflammation [111].

The alga *Spirulina* had demonstrated anti-inflammatory effects when assessed using a non-alcoholic steatohepatitis model [112]. C-phycocyanin from *Spirulina platensis* blocked inflammation via inhibiting the expressions of nitric oxide synthase, cyclooxygenase-2, and production of pro-inflammatory cytokines [113,114]. Methanolic extracts of *Ulva lactuca* and *U. conglobate* have shown anti-inflammatory effects in murine hippocampal HT22 cell line [115]. Lycopene from *Chlorella marina* demonstrated anti-inflammatory effects in an arthritic rat model [116]. Phytosterols from *Dunaliella tertiolecta,* aqueous and methanolic extracts of *Caulerpa mexicana* and lectin from *Caulerpa cupressoides* exhibited anti-inflammatory activities in several in vitro models [117]. Ethanolic extract of *Ecklonia cava* inhibited LPS-induced inflammation in human endothelial cells [118]. Furthermore, *Ishige okamurae* showed anti-inflammatory effects in a few in vitro models [119]. The astaxanthin isolated from *Haematococcus pluvialis* reduced gastric inflammation in *Helicobacter pylori*-infected mice via decreasing bacterial density [120]. Moreover, astaxanthin reduced the production of pro-inflammatory mediators and cytokines such as nuclear factor-κB (NF-κB), tumor necrosis factor-α (TNF-α), and interleukin-6 (IL-6), and suppresses T lymphocyte activation in asthma patients [120]. Fucans from *Sargassum vulgare, Lobophora variegata,* and *Spatoglossum schroederi* also displayed anti-inflammatory effects [121]. Furthermore, Alginic acid from *Sargassum wightii* exhibited anti-inflammatory effects in vivo in a rat model [121].

Methanolic extract of *Bryothamnion triquetrum* exhibited an anti-inflammatory effect in Swiss albino mice [122]. Two fatty acids of *Gracilaria verrucosa* such as (*E*)-10-oxooctadec-8-enoic acid and (*E*)-9-oxooctadec-10-enoic acid inhibited the production of inflammatory markers, such as nitric oxide, IL-6, and TNF-α [123]. The sulfoglycolipidic isolated from the red alga *Porphyridium cruentum* exhibited an anti-inflammatory effect due to the presence of AA (6.8%), palmitic acid (26.1%), and EPA (16.6%), and omega-9 fatty acid (10.5%) [124]. Pheophytin from *Enteromorpha prolifera* has superoxide radical (O_2_^•_^) reducing potential and inflammatory responses in mice [125,126]. A glycoprotein extracted from *Porphyra yezoensis* exhibited anti-inflammatory effects in LPS-stimulated macrophages [118]. Furthermore, phlorotannins (a polyphenol derived from *Eisenia bicyclis, Ecklonia cava*, and *Ecklonia kurome*), and sargachromanol G (derived from Sargassum siliquastrum) showed promising anti-inflammatory activity via inhibition of the production of inflammatory mediators in LPS-stimulated cells [127,128]. Moreover, methanolic extract of *Neorhodomela aculeata* inhibited ROS generation, H_2_O_2_-induced lipid peroxidation, and inducible nitric oxide synthase in neurological diseases via inhibition of inflammation [129].

### 3.4. Significance of Marine Algal Bioactive Metabolites as Anti-Diabetes Drugs

Diabetes mellitus is a chronic metabolic disorder which is characterized by high blood glucose levels that lead to renal dysfunction, cardiovascular diseases, and retinal damage [1]. Dietary management is a novel target for treating diabetes via maintaining the correct concentrations of both blood glucose and blood lipids [130]. Commercially available antidiabetic drugs exert several diseases-associated adverse side effects during treatments [131]. In this context, the identification of natural antidiabetic drugs with enhanced drug efficacy and lesser adverse effects has gained the attention of researchers in recent times. Marine algae-derived bioactive compounds exhibited antidiabetic properties furnished by regulation of various signaling pathways, such as inhibitory effect on enzymes such as α-amylase, α-glucosidase, aldose reductase, dipeptidyl peptidase-4, and protein tyrosine phosphatase 1B (PTP 1B) enzyme [132]. Enzymes like α-amylase and α-glucosidase play a significant role in the digestion of carbohydrates, leading to a delay in glucose absorption in blood and also to a reduction of glucose levels in blood plasma. Subsequently, these compounds may be exploited as potential functional food ingredients for preventing or diminishing insulin resistance and diabetes [132].

Marine algal compounds modulated the GLUT-4 and AMPK signaling pathways and triggered glucose tolerance [1]. Recent investigations displayed that fucoidan act as prebiotics and regulate the intercellular metabolism and blood sugar level [1]. Fucoidan isolated from *S. fusiforme* controlled the blood glucose level, recovered liver function, and inhibited oxidative stress in STZ-induced diabetic rats [133]. Fucoidan from *Ecklonia maxima* acted as a potent α-glucosidase inhibitor with a very low IC_50_ value (0.27–0.31 mg/mL) and regulated type II diabetes [134]. Fucoidan from *Fucus vesiculosus* displayed a robust α-glucosidase inhibitor in diabetes treatment [135]. Furthermore, low molecular weight fucoidan (LMWF) from *S. hemiphyllum* in combination with fucoxanthin displayed anti-diabetic properties in type II diabetes rat model (db/db). The oral administration of LMWF in combination with fucoxanthin decreased blood glucose and fasting blood sugar levels [135]. The synergistic drug effect was more effective in the in vivo model via reduction of urinal sugar level as compared to the LMWF treatment alone. LMWF enhanced the hepatic glycogen concentration and antioxidant enzymes which were assisted by lipid metabolism. The lipid metabolism displayed the regulation of glucose transporter (GLUT), insulin receptor substrate (IRS-1), peroxisome proliferator-activated receptor-gamma (PPARγ), and uncoupling protein (UCP)-1 level with the treatment of LMWF in combination with fucoxanthin [135]. Fucoidan from *Cucumaria frondosa* amplified the expression of insulin receptor substrate 1, Glut-4, and PI3K/Akt, glucose transporter protein in insulin-resistant rats [1]. Fucoidan from *Saccarina japonica* abridged blood sugar level too [1,136].

Moreover, sulfated fucoidan isolated from *Undaria pinnatifida* inhibited hyperglycemia by eliciting insulin sensitivity in a diabetic mouse (C57BL/KSJ/db/db) model [137]. Fucoidan from *Sargassum wightii* inhibited alpha-D-glucosidase that transport glucose into the blood and reduce glucose level in blood [138]. Dieckol, fucodiphloroethol G, 6,6′-Bieckol, 7-phloroeckol, phlorofucofuroeckol A from *E. cava* and phloroglucinol, dioxinodehydroeckol, eckol from *E. stolonifera* and *E. bicyclis* displayed robust α-glucosidase activity and reduced blood sugar level [139,140,141]. Furthermore, dieckol-rich extract from *E. cava* improved insulin sensitivity [142]. Polyphenolic-rich extract from *I. okamurae* improved insulin sensitivity [143]. Moreover, polyphenolic-rich extract from *E. cava* inhibited glucose uptake effect in skeletal muscle [141]. Fucosterol from *Pelvetia siliquosa* reduced serum glucose concentration and inhibited sorbitol accumulation in the lenses in Sprague–Dawley diabetic rats [144]. Phlorotannin components from *Ascophyllum nodosum* displayed potential inhibition of α-amylase and α-glucosidase activities in in vitro models [145]. Sodium alginate from *Laminaria angustata* inhibited the rising blood glucose and insulin levels in Wistar rats [146]. Fucoxanthin and fucosterol from *Undaria pinnatifida* and *Ecklonia stolonifera* displayed aldose reductase inhibition [147,148]. Furthermore, pheophorbide-A, pheophytin-A also displayed aldose reductase inhibition [149]. Algal bioactive metabolites and their functional role in diabetes are summarized in Table 2.

## 4. Algal Metabolites as Prebiotics for Human Health with Special References to Fucoidan

Algal metabolite consumption, such as polysaccharides, sulfated polysaccharides, fucoidans, chlorophylls, phycobilins, fucoxanthins, carotenoids, polyphenols, and omega-3 fatty acids, decreases blood pressure and sugar level, and it can have antiviral, anti-inflammatory, anticancer, and neuroprotective effects, as well as act as immune boost-up. The immunomodulatory potential of prebiotics modulates immune fitness via several metabolic processes and interactions with the gut microbiota in humans [165]. The gut microbiota produces short-chain fatty acids (SCFA) such as propionate, acetate and butyrate by breaking down prebiotics and modulating the immune response [165]. Intravenous use of acetate amplified the activity of NK cells in cancer patients. In addition, it activated G protein-coupled receptors (GPR41 and GPR43) in rats, thus triggering mitogen-activated protein kinase (MAPK) signaling and modulating the transcription factors activity [166]. Acetate also increased the production of IL-10 in rats and prevented the inhibitory activity of butyrate on IL-2 production [167]. Daily administration of fucoidans from *A. nodosum* increased in *Lactobacillus* and *Ruminococcus* in the intestine of mice [168].

Fucoidan possesses a wide range of immune-modulation effects by stimulating activation of natural killer (NK) cells, dendritic cells (DCs), and T cells and increasing anti-tumor and anti-viral responses [169]. Fucoidan enhanced immune modulation via activation of macrophage facilitated by membrane receptors, such as TLR4, cluster of differentiation 14 (CD14), competent receptor-3 (CR-3), and scavenging receptor (SR). This led to signal transduction via MAPK and activation of transcription factors, and it also induced cytokines production, which regulates activation of NK cells and T lymphocytes [170]. In this regard, treatment of C57BL/6 rats with fucoidan extracted from *Fucus vesiculosus* up-regulated pro-inflammatory cytokines (IL-6, IL-12, and TNF-α) in serum and spleenocytes after 3 h of administration [171]. Furthermore, fucoidan from *L. cichorioides*, *L. Japonica*, and *F. evanescens* served as TLR ligands and their interaction with TLR-2 and TLR-4 receptors in vitro activated NF-jB. Furthermore, it also controlled the expression of the defense mechanisms of intrinsic immunity, such as secretion of chemokines, cytokines, and manifestation of MHC class I and II particles [1]. These are essential for the defense against foreign attackers and for activating adaptive immune systems. A clinical trial based on the diet supplementation of 1 g/day of fucoidan from *Undaria pinnatifida* on adult male and female volunteers for 24 weeks showed modulation of the immunity to seasonal influenza vaccine by antibody production [172]. Based on these reports, there are several evidences that fucoidan acts as a potent prebiotic and that it is able to modulate immunity. This is achieved by interacting with intestinal cells of the gut microbiota and direct motivation of immune cells through TLRs.

## 5. Conclusions and Future Perspectives

Natural extracts have been used since ancient times for treating various illnesses. Products from natural products have also provided a large number of pharmaceuticals, or their prototypes, in recent times. Among the many natural sources, marine algae can still play a pivotal role in human health and disease because of the need for novel drug candidates. Their extracts are already well-known in traditional medicine and more recent studies investigated the many beneficial effects of their secondary metabolites, such as reduction of oxidative stress and modulation of apoptosis, and the main findings of these researches are summarized in this review. The exploitation of these results might lead to the development of novel algal dietary supplements and pharmaceuticals for preventing and treating chronic malfunctions and other age-associated chronic diseases. For example, sulfated fucoidans have been shown to be potential candidates as new pharmaceuticals in fighting cancer and diabetes. However, despite the extensive use of algae-derived compounds and extracts in the food industry, there are still no FDA-approved anticancer, antioxidant, anti-inflammatory, and antidiabetic drugs. Technology transfer from the pre-clinical results to the clinical application of secondary metabolites extracted from marine algae is still in its infancy and not fully exploited and more clinical studies are needed to really evaluate the pharmaceutical efficacy algal compounds.

In conclusion, marine algae offer a great variety of bioactive molecules with potential health benefits. Several types of marine algae are already consumed as food additives and nutritional supplements, potentially exerting their beneficial effects through diet. There is, however, an impelling necessity of considering the algal bioactive compounds in new drug discovery programs and to investigate their biological effects in deeper detail in order to find new pharmaceuticals with preventive and therapeutic efficacy.

## Figures and Tables

**Figure 1 molecules-26-00037-f001:**
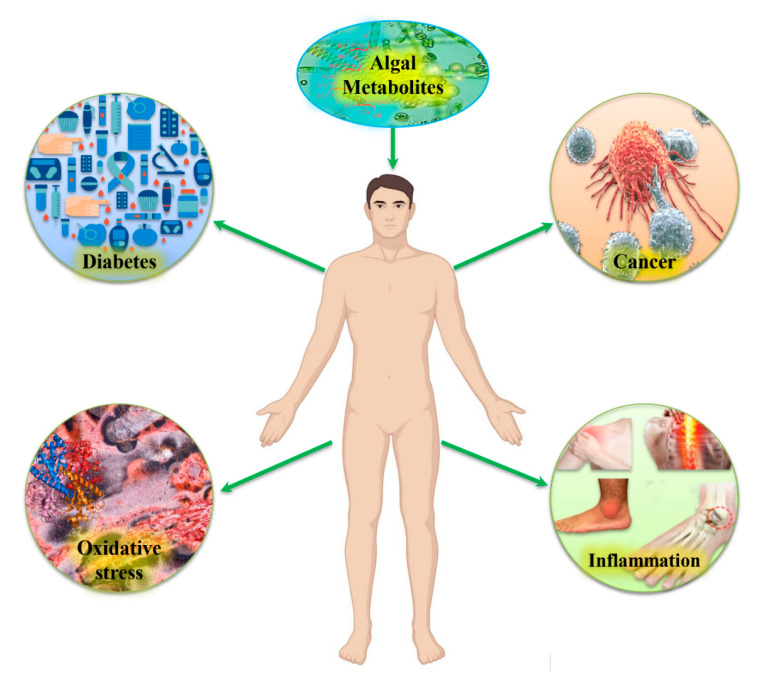
Potential beneficial effects of algal metabolites on human health. Secondary metabolites in marine algae could provide novel drug candidates for fighting various diseases, e.g., by reducing the α-amylase and α-glucosidase activity in diabetes; by reducing inflammation thanks to their antioxidant capacity; and inhibiting cellular proliferation in tumor cells.

**Figure 2 molecules-26-00037-f002:**
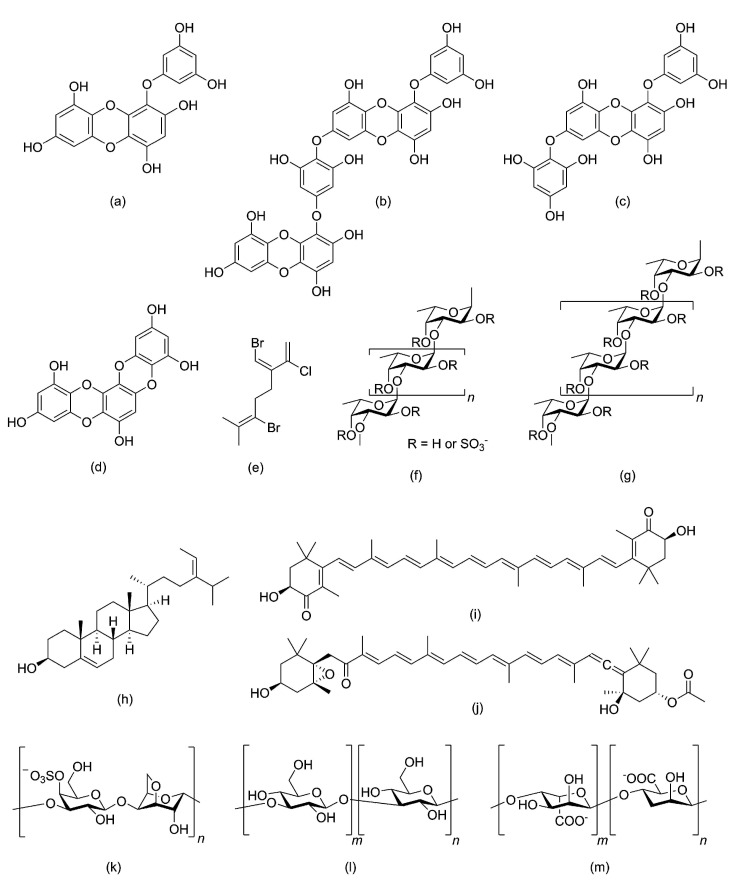
Chemical structure of several algal metabolites that can have beneficial health effects by acting as antioxidants: (**a**) eckol, (**b**) dieckol, (**c**) 7-phloroeckol, and (**d**) dioxinodehydroeckol, four phlorotannins; (**e**) 2-chloro-3-(bromomethylene)-6-bromo-7-methyl-1,6-octadiene, a halogenated monoterpene; (**f**) type I and (**g**) type II fucoidans; (**h**) fucosterol; (**i**) astaxanthin and (**j**) fucoxanthin; (**k**) k-carrageenan; (**l**) laminaran; and (**m**) alginate.

**Figure 3 molecules-26-00037-f003:**
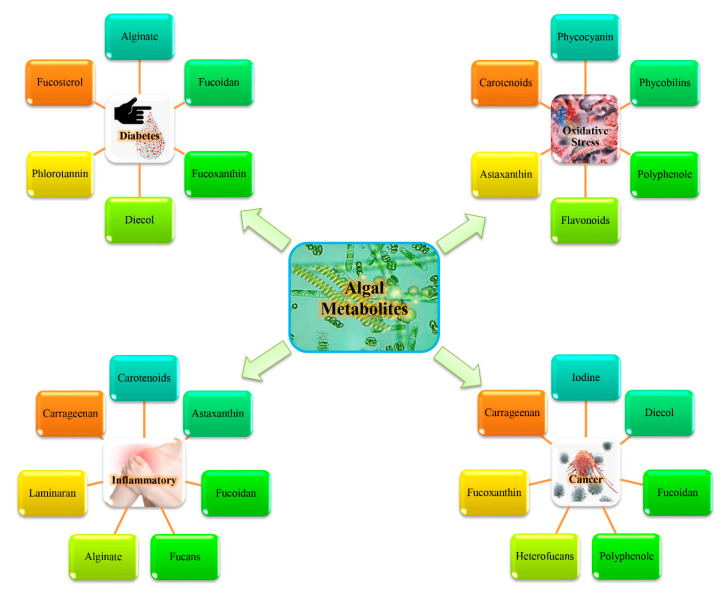
Potential effects of algal metabolites in different human disease.

**Figure 4 molecules-26-00037-f004:**
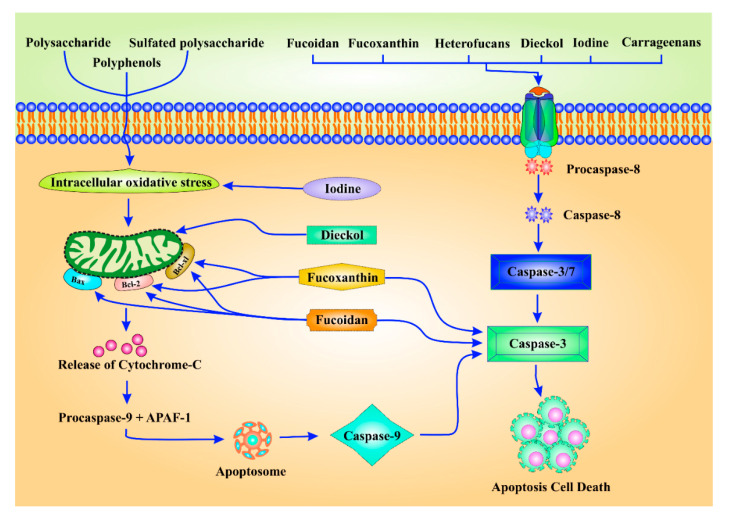
Apoptosis modulation by algal metabolites in cancer prevention. Polysaccharides, sulfated polysaccharides, iodine, dieckol, fucoxanthine, fucoidan, and polyphenols downregulate the expression of anti-apoptotic protein Bcl-xl, Bcl-2. Similarly, they enhance the Bax expression to aid apoptosis. Fucoidan supports the intrinsic apoptosis via regulating the cytosolic release of cytochrome C. Fucoxanthine and fucoidan induces the expression of caspase 9 and caspase 3 to induce apoptotic cell death. Moreover, fucoidan, fucoxanthine, heterofucans, dieckol, iodine, and carrageenans induce apoptosis through modulation of caspase 3 activity via death receptor-mediated apoptotic cell death in several cancer cell lines. In addition to this, they also regulate caspase 8 activity inducing extrinsic apoptosis in different cancer cells.

**Table 1 molecules-26-00037-t001:** Marine algal bioactive metabolites and their functional role in apoptosis.

	Bioactive Compounds	Algal Sources	Cell Lines/In Vivo Models Involved	Functional Involvement	Ref.
1	Sulfated polysaccharides	Brawn algae	Human leukemic monocyte lymphoma cell line (U-937)	Inhibition of cell proliferation	[1]
2	Polysaccharides	*Capsosiphon fulvescens*	Gastric cancer cells	Modulation of PI3K/Akt pathway	[96]
3	Polysaccharides	*U. lactuca*	DMBA-induced breast cancer model	Diminished lipid peroxidation also GPx activity	[97]
4	Polysaccharides	*Champia feldmannii*	Mice transplanted with sarcoma 180 tumors	Reduction of tumor growth	[98]
5	Polysaccharides	*U. lactuca*	DMBA-induced breast cancer in rat	Prevented breast-histological alterations and carcinogenic wounds	[97]
6	Polysaccharides	*U. lactuca*	Breast cancer cells	p53 expression and inhibited the Bcl-2 expression	[97]
7	Fucoidan	*Costaria costata*	DLD-1	55% (100 μg/mL)	[107]
			SK-MEL-28	20% (100 μg/mL)	[107]
8	Fucoidans	*U. pinnatifida*	DU-145 cell-induced xenograft rat model	Inhibition of the JAK3/STAT pathway	[99]
9	Fucoidan	*Fucus vesiculosus*	HCT116 human colorectal carcinoma cell line	p53-independent	[74]
10	Fucoidan	*L. japonica*	HT-29 cell line	Caspase-3, PARP, and DNA degradation	[100]
11	Fucoidan	*E. cava*	MDA-MB231 and MCF-7 cells	Induction of p53 and activation of Bax, caspases 3 and 9, and PARP with inhibition of Bcl-2	[76]
12	Fucoidan	*Cladosiphon okamuranus*	MCF-7 cells	Activation of caspase-3 and DNA fragmentation	[101]
13	Extracts	*F. vesiculosus*	Human colon cancer cells	Cytoplasmic release of cytochrome C and the Smac/DIABLO pathway	[102]
14	Fucoidan	*F. vesiculosus*	HT-29 colon cancer cells	Decreased expression of Bcl-xL, Bcl-2 and upregulation of Bax, pro-caspases 3, 7, and 9	[103]
15	Fucoidan	*F. vesiculosus*	HT-29 colon cancer cells	Upregulation of Rb and E2 factor proteins and Fas regulation	[103]
16	Fucoxanthin	*Laminaria japonica*	Human colon adenocarcinoma WiDr cells	DNA fragmentation	[91]
17	Fucoxanthin	*Ishige okamurae*	Melanoma B16F10 cells	Caspases activation and reduction of BclxL and IAP expression	[90]
18	Dieckol	*E. cava*	Albino rats	Decreased cancer cell proliferation	[95]
19	Guaiane sesquiterpene	*Ulva fasciata*	MDA MB-231 breast cancer cell line	Direct interaction with kinase site of EGFR	[82]
20	Halogenated monoterpene, mertensene	*Pterocladiella capillacea*	HT-29 and LS174	ERK1/2, Akt, and NF B pathways	[78]
21	Ethanolic and methanolic extracts	*Gracilaria tenuistipitata*	Ca9-22 oral cancer cells	DNA damage	[88]
22	Methanolic extract	*Plocamium telfairiae*	HT-29 colon cancer cells	Induced caspase-dependent	[88]
23	Iodine and polyphenols	*L. japonica*	HT-29 colon cancer cells	Inhibition of SOD activity	[105]
24	Extract	*Eucheuma cottonii*	Cancer-induced rats	Upregulation of antioxidant enzymes, e.g., CAT, SOD, and GPx	[89]
25	Methanolic extracts	*Fucus serratus* and *F. vesiculosus*	Caco-2 cells	DNA damage	[106]
26	Methanolic extracts	*Pelvetia canaliculata*	Caco-2 cells	Inhibited H_2_O_2_-induced SOD depletion	[106]
27	Ethanol extracts	*C. japonica*	Caco-2 cells	Activating cellular antioxidant enzymes	[63]
28	Aqueous extract	*G. tenuistipitata*	H1299 cell line	Activating cellular antioxidant enzymes	[59]

**Table 2 molecules-26-00037-t002:** Marine algal bioactive metabolites and their functional role in diabetes.

	Algal Type	Algal Sources	Bioactive Metabolites	Functional Involvement	Ref.
1	Red algae	*Rhodomela confervoides*	HPN analogues	Inhibition of PTP 1B	[150]
2	Brown alga	*Ecklonia cava*	Methanolic extract	Increases phosphorylation AMP-activated protein kinase; radical scavenging property	[151]
3		*Isochrysis galbana, Nannochloropsis oculata*	DHA, EPA	Regulates glucose and lipid metabolism	[152]
4		*Palmaria, Ascophyllum, Alaria*	Phenol rich extract	Inhibitory of α-amylase and α-glucosidase	[145]
5	Brown algae		Polyphenols/Phlorotannins	Inhibition of α-glucosidase and α-amylase; increases skeletal muscle glucose uptake; inhibition of PTP 1B enzyme; increases insulin sensitivity.	[153]
6	Brown algae	*E. cava*	Dieckol	α-Glucosidase inhibitor [20]; postprandial hyperglycemia-lowering effect [7]; glucose uptake effect in skeletal muscle [40]; PTP 1B inhibition [10]; protective effect against diabetes complication	[139,140,154,155]
7	Brown algae	*E. cava*	Fucodiphloroethol G	α-Glucosidase inhibitor	[140]
8	Brown algae	*E. cava*	6,6′-Bieckol	α-Glucosidase inhibitor	[140]
9	Brown algae	*E. cava*	7-Phloroeckol	α-Glucosidase inhibitor; PTP 1B inhibition	[139,140]
10	Brown algae	*E. cava*	Phlorofucofuroeckol A	α-Glucosidase inhibitor; PTP 1B inhibition	[139,140]
11	Brown algae	*E. stolonifera* *E. bicyclis*	Phloroglucinol	α-Glucosidase inhibitor; PTP 1B inhibition	[139]
12	Brown algae	*E. stolonifera* *E. bicyclis*	Dioxinodehydroeckol	α-Glucosidase inhibitor; PTP 1B inhibition	[139]
13	Brown algae	*I. okamurae*	Diphlorethohydroxycarmalol	α-Glucosidase inhibitor; postprandial hyperglycemia-lowering effect; protective effect against diabetes complication	[156]
14	Brown algae	*E. stolonifera* *E. bicyclis*	Eckol	α-Glucosidase inhibitor; PTP 1B inhibition	[139]
15	Brown algae	*I. foliacea*	Octaphlorethol A	Glucose uptake effect in skeletal muscle	[142]
16	Brown algae	*A. nodosum* *F. vesiculosus*	Polyphenolic-rich extract	α-Glucosidase inhibitor; postprandial hyperglycemia-lowering effect	[145]
17	Brown algae	*E. cava*	Polyphenolic-rich extract	Glucose uptake effect in skeletal muscle	[141]
18	Brown algae	*E. cava*	Dieckol-rich extract	Improvement of insulin sensitivity	[142]
19	Brown algae	*I. okamurae*	Polyphenolic-rich extract	Improvement of insulin sensitivity	[143]
20		*Ulva rigida*	Ethanolic extract	Decreased blood glucose concentrations in Wistar diabetic rats	[157]
21		*Pelvetia siliquosa*	Fucosterol	Reduction of serum glucose levels and inhibition of sorbitol accumulation in the lenses in Sprague–Dawley diabetic rats	[144]
22	Brown algae	*Ecklonia cava*	Methanolic extract	Reduction in plasma glucose levels and increased insulin concentration; activation of AMPK/ACC and PI3/Akt signaling pathways in Sprague–Dawley diabetic rats	[141]
23	Green algae and Diatoms	*Chlorella* sp., *Nitzschia laevis*	Microalgal extracts	Inhibition of advanced glycation endproducts (AGEs) formation in in vitro models	[158]
24	Brown algae	*Ascophyllum nodosum*	Phlorotannin components	Inhibition of α-amylase and α-glucosidase activities in vitro	[145]
25	Brown algae	*Laminaria angustata*	Sodium alginate	Inhibition of rising blood glucose and insulin levels in Wistar rats	[146]
26	Brown algae	*Eisenia bicyclis*	Dioxinodehydroeckol	α-Glucosidase inhibitor	[159]
27	Brown algae	*Eisenia bicyclis*	7-Phloroeckol	PTP 1B inhibition; α-glucosidase inhibitor	[159]
28	Brown algae	*Eisenia bicyclis*	Fucoxanthin	Aldose reductase inhibition	[148]
29	Brown algae	*Ecklonia cava*	Dieckol	α-Glucosidase inhibitor	[160]
30	Brown algae	*Ecklonia cava*	Fucodiphloroethol G	α-Amylase inhibitor	[160]
31	Brown algae	*Ecklonia cava*	6,6′-Bieckol	PTP 1B inhibition	[160]
32	Brown algae	*Ecklonia cava*	7-Phloroeckol	ACE inhibitor	[160]
33	Brown algae	*Ecklonia cava*	2-Phloroeckol	α-Glucosidase inhibitor; α-glucosidase inhibitor; α-glucosidase inhibitor; PTP 1B inhibition; aldose reductase inhibition; aldose reductase inhibition	[139,161]
34	Brown algae	*Ecklonia cava*	Phlorofucofuroeckol A	α-glucosidase inhibitor; PTP 1B inhibition; ACE inhibitor; AGEs inhibition; Aldose reductase inhibition	[139,162]
35	Brown algae	*Ecklonia stolonifera*	Phloroglucinol	α-glucosidase inhibitor	[153]
36	Brown algae	*Ecklonia stolonifera*	Eckol	PTP 1B inhibition α-glucosidase inhibitor	[153]
37	Brown algae	*Ecklonia stolonifera*	Dieckol	α-Amylase inhibitor; ACE inhibitor; PTP 1B inhibition; α-glucosidase inhibitor	[144,160]
38	Brown algae	*Ecklonia stolonifera*	Phlorofucofuroeckol A	α-Amylase inhibitor; PTP 1B inhibition; ACE inhibitor; α-glucosidase inhibitor	[144,160]
39	Brown algae	*Ecklonia stolonifera*	Fucosterol	PTP 1B inhibition; aldose reductase inhibition	[147]
40	Brown algae	*Ishige okamurae*	Diphlorethohydroxycarmalol	α-Glucosidase inhibitor; α-amylase inhibitor	[163]
41	Brown algae	*Myagropsis myagroides*	Eckol	α-Glucosidase inhibitor; α-amylase inhibitor	[153]
42	Brown algae	*Sargassum serratifolium*	Sargahydroquinoic acid	ACE inhibitor; PTP 1B inhibition	[164]
43	Brown algae	*Ascophyllum nodosum*	Methanol extract	α-Glucosidase inhibitor; α-amylase inhibitor	[145]
44	Brown algae	*Saccharina japonica*	Pheophorbide-A	Aldose reductase inhibition	[149]
45	Brown algae	*Saccharina japonica*	Pheophytin-A	Aldose reductase inhibition	[149]
46	Brown algae	*Undaria pinnatifida*	Fucoxanthin	Aldose reductase inhibition	[148]

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
