# Peer review of "Bioactive Metabolites from Marine Algae as Potent Pharmacophores against Oxidative Stress-Associated Human Diseases: A Comprehensive Review"

_molecules, 2020, doi:10.3390/molecules26010037_

Round 1

Reviewer 1 Report

During the review, I found the human diseases of bioactive metabolite a a very interesting one.

In terms of metabolite structure, suggest adding a representative structure of bioactive compounds as Figure.  for example fucoidan, fucoxanthin, dieckol, HPN, DHA, EPA, ....

Author Response

During the review, I found the human diseases of bioactive metabolite a a very interesting one.

We thank the referee for the kind comment and recognition of the hard work behind this article.

In terms of metabolite structure, suggest adding a representative structure of bioactive compounds as Figure.  for example fucoidan, fucoxanthin, dieckol, HPN, DHA, EPA, ....

We thank the referee for the constructive comment. As suggested, we included a new figure with the chemical structures of several representative secondary metabolites from marine algae.

Reviewer 2 Report

The review has a great interest. Algae is a very unexplored source of nutrients and bioactive compounds and the extension of it use could help to resolve some of the main challenges facing humanity with climate change and the exponential growth of the world's population. The review is well-written and only minor changes are needed.

Line 64: Algae extracts have showed also antimicrobial activity and its possible use as a natural alternative to preservatives. This information can be added to the manuscript (i.e):

-Effect of an icing medium containing the alga Fucus spiralis on the microbiological activity and lipid oxidation in chilled megrim (Lepidorhombus whiffiagonis).

-Impact of icing systems with aqueous, ethanolic and ethanolic-aqueous extracts of alga Fucus spiralis on microbial and biochemical quality of chilled hake (Merluccius merluccius).

Line 141: To the best of my knowledge, starch is also a polysaccharide. The subdivision of this section is a bit confusing. Please joint in one section and rename.

Maybe the authors could add a figure with the main component of the different algae an its potential benefic for the health. This should help to understand the role of each macroalgae component in human health.

Also the authors could add some information of the prebiotic effect of algae and its influence in human health (i.e: Potential use of marine seaweeds as prebiotics: A review)

Author Response

The review has a great interest. Algae is a very unexplored source of nutrients and bioactive compounds and the extension of it use could help to resolve some of the main challenges facing humanity with climate change and the exponential growth of the world's population. The review is well-written and only minor changes are needed.

We thank the referee for kind comment and for the recognizing its importance to a wide audience.

Line 64: Algae extracts have showed also antimicrobial activity and its possible use as a natural alternative to preservatives. This information can be added to the manuscript (i.e):

-Effect of an icing medium containing the alga Fucus spiralis on the microbiological activity and lipid oxidation in chilled megrim (Lepidorhombus whiffiagonis).

-Impact of icing systems with aqueous, ethanolic and ethanolic-aqueous extracts of alga Fucus spiralis on microbial and biochemical quality of chilled hake (Merluccius merluccius).

We thank the referee for the constructive comment. As requested, we discussed the antimicrobial activity of marine algae and included the suggested references (lines 66-68).

Line 141: To the best of my knowledge, starch is also a polysaccharide. The subdivision of this section is a bit confusing. Please joint in one section and rename.

We thank the referee for the suggestion, we jointed the two sections (now section 2.3).

Maybe the authors could add a figure with the main component of the different algae an its potential benefic for the health. This should help to understand the role of each macroalgae component in human health.

We thank the referee for the constructive comments. We have added two more figures, one (Figure 2) with the chemical structures of several representative active metabolites from marine algae and another (Figure 3) with their potential beneficial effects.

Also the authors could add some information of the prebiotic effect of algae and its influence in human health (i.e: Potential use of marine seaweeds as prebiotics: A review)

We thank the referee for the nice suggestion, we added a paragraph (Section 4) about the prebiotic effect of marine algae and included the suggested reference (lines 478-510).